# The Fast Evolution of the Stenobothrini Grasshoppers (Orthoptera, Acrididae, and Gomphocerinae) Revealed by an Analysis of the Control Region of mtDNA, with an Emphasis on the *Stenobothrus eurasius* Group

**DOI:** 10.3390/insects15080592

**Published:** 2024-08-03

**Authors:** Svetlana Sorokina, Nikita Sevastianov, Tatiana Tarasova, Varvara Vedenina

**Affiliations:** 1Koltzov Institute of Developmental Biology, Russian Academy of Sciences, 26 Vavilov Street, Moscow 119334, Russia; svetlana_ibr@mail.ru; 2Institute for Information Transmission Problems, Russian Academy of Sciences, 19 Bolshoy Karetny per., Moscow 127051, Russia; tegipokorolet@gmail.com (N.S.); thomisida@gmail.com (T.T.)

**Keywords:** acoustic communication, divergence rate, phylogeny, contact zone, glacial refugia

## Abstract

**Simple Summary:**

Grasshopper species of the Stenobothrini tribe are known by the wide diversity of their premating traits, such as acoustic behavior and morphological characters. However, the phylogenetic relationships, divergence time, and phylogeography of these species inferred with the mtDNA markers (the control region and the COI gene) did not reflect the species divergence. The analysis of interspecific and intraspecific variability in the control region in two hybridizing species of this genus, *S. eurasius* and *S. hyalosuperficies*, supports the hypothesis of putative hybridization between the species and shows the direction of the mtDNA introgression.

**Abstract:**

The two cryptic grasshopper species of the genus *Stenobothrus*, *S. eurasius* and *S. hyalosuperficies*, demonstrate different acoustic behavior despite a strong similarity in morphology. A hybridization between these species is possible in the contact zone; however, there are little molecular data about the relationships of these species. The analysis of the mtDNA control region (CR) reveals that haplotypes of *S. hyalosuperficies* have more in common with the more distant *Stenobothrus* species than with the closely related *S. eurasius*. In the contact zone, *S. eurasius* has mt-haplotypes shared with *S. hyalosuperficies*, which might indicate an introgression of mtDNA from *S. hyalosuperficies* to the *S. eurasius* gene pool. We also analyze the structure and evolutionary rate of the mtDNA CR for the *Stenobothrus* genus and estimate the time of divergence of the species within the genus. The phylogenetic tree of the tribe Stenobothrini reconstructed with either the CR or COI gave the same four groups. The phylogenetic tree of the *Stenobothrus* genus has a star-like topology with each mtDNA haplotype found in any analyzed species, except *S. eurasius*, which forms a separate branch. The maximum degree of incomplete lineage sorting can demonstrate either ancestral polymorphism or introgression.

## 1. Introduction

The ways of establishing reproductive isolation between species have traditionally received much attention. Interspecific reproductive barriers can be classified into premating and post-mating (prezygotic and post-zygotic) isolation mechanisms according to the time when they occur during the life cycle [1,2,3]. Intrinsic post-zygotic incompatibilities such as hybrid unviability and sterility have been considered as the classical driving force behind speciation [3]. At the same time, more intriguing processes occur when premating isolation evolves faster than post-zygotic barriers. It is assumed that premating isolation may be achieved by assortative mating evolving as a result of sexual selection, when a parallel change in mate preference and secondary sexual traits takes place [4,5,6].

In many species of animals, acoustic communication is one of the important components of premating isolation. Among animal groups that exhibit acoustic communication, grasshoppers of the subfamily Gomphocerinae are considered to be a good model for the studies of rapid speciation. First, there are many cryptic species that differ only by songs (e.g., [7,8,9,10]). Second, among Orthoptera, acoustic communication in Gomphocerinae is most developed in terms of the complexity of the sound-producing mechanisms, the number of sound elements, and the mating strategies. In particular, the majority of the gomphocerine grasshoppers produce two song types, the calling song and courtship song, and the courtship songs may reach a high degree of complexity and be accompanied by visual signals (e.g., [7,9,11,12]). Finally, many gomphocerine species hybridize in nature (e.g., [13,14,15]), which is indicative of a recent and non-complete divergence of these young species.

The two cryptic species of the genus *Stenobothrus*, *S. eurasius* and *S. hyalosuperficies*, represent an interesting group within Gomphocerinae. Despite a strong similarity in morphology, they are different both in their calling and courtship songs and in their sound-producing mechanisms [10]. *S. eurasius* generates sound by the common tegminal–femoral stridulation, whereas *S. hyalosuperficies* generates sound by both common stridulation and wing clapping. The range of *S. eurasius* extends from south-eastern Europe to southern Siberia, eastward to Transbaikalia; the range of *S. hyalosuperficies* is smaller, covering the south-eastern part of European Russia, the Orenburg region, Kazakhstan except the northern regions, Kyrgyzstan, and north-western China. Despite the striking differences in songs and sound-producing mechanisms between the two species, they were found to hybridize in the wide contact zone [16]. We also conducted pilot behavioral experiments with the song playback, which showed that the females of *S. hyalosuperficies* are less selective than the females of *S. eurasius* [16].

These preliminary results do not allow us to unambiguously determine the phylogenetic relationships between the two species. On one hand, a phylogenetic analysis of Gomphocerinae showed that all extant species with ancient origins produce songs only by means of stridulation (e.g., [11,17]). Another sound-producing mechanism, the crepitation, is known for a relatively small number of species, mainly within the genus *Stenobothrus*, which is suggested to be a group of young species. The divergence of this genus is estimated from 0.6 mya [18] to 1.9 mya [17]. Thus, we propose that *S. eurasius* is closer to an ancestral form than *S. hyalosuperficies.* On the other hand, the lower selectivity of the *S. hyalosuperficies* females found in the behavioral experiments might indicate that this species is closer to an ancestral form than *S. eurasius*.

It was shown that interspecific boundaries within the cryptic species complexes can be rarely identified by using a standard set of genetic markers (e.g., [19]). A low mutation rate and an ancestral polymorphism make the molecular differentiation of taxa difficult. The previously obtained phylogenetic reconstructions of Gomphocerinae demonstrate the extremely small genetic distances between haplotypes of closely related species and do not reveal the expected clustering [11,17,20,21]. We compared the values of pairwise interspecific genetic distances for the tribe Stenobothrini for each common mitochondrial marker and found the mtDNA control region (CR) to be the most variable. Unlike the mt-genomes of some other Orthoptera species with a large CR (e.g., Podismini; [22]), the mt-genomes of the Stenobothrini species have a relatively short CR with a length of about 770 bp [23,24]. Moreover, the sequence of the Stenobothrini CR is poorly enriched in A and T compared to the *Drosophila* sequences [25,26] and does not contain Variable Number Tandem Repeats (VNTRs), which are found in the CR of other groups of Orthoptera species (e.g., Gryllotalpidae; [27]). The CR does not belong to the mt-genome part, which is overrepresented in the nuclear genome as NUMTs [28,29]. All of the above shows the CR of mtDNA to be a promising marker for the study of divergence and the phylogeographic history of the species *S. eurasius* and *S. hyalosuperficies*.

In the current study, we analyze the structure and evolutionary rate of the mtDNA CR for the genus *Stenobothrus* and trace the phylogenetic relationships between species of the Stenobothrini tribe based on the analysis of the CR and the COI gene. We estimate the time of divergence of the species within the genus *Stenobothrus* and discuss the phylogeography of this genus. In the two closely related species, *S. eurasius* and *S. hyalosuperficies*, the interspecific and intraspecific variability in the CR is analyzed. Following up on our results, we test the following three hypotheses: (1) if the phylogenetic tree shows *S. eurasius* as paraphyletic with regard to *S. hyalosuperficies*, this would indicate that *S. hyalosuperficies* diverged from *S. eurasius* or from a common ancestor; (2) if the phylogenetic tree shows a reciprocal monophyly, this would be evidence of the existence of periods of population decline and the subsequent expansion of both species accompanied by the formation of mt-haplogroups; (3) and finally, a polyphyletic structure of the phylogenetic tree would indicate either the common gene pool of the two species due to ancestral polymorphism or an insufficient resolution of the marker used. A comparative analysis of the haplotypes from allopatric populations and the contact zone between *S. eurasius* and *S. hyalosuperficies* would allow us to support putative hybridization and to show the direction of the mtDNA introgression.

## 2. Material and Methods

### 2.1. Sample Collection and DNA Extraction

During 2022–2023, 82 specimens of 17 species of the Stenobothrini tribe and the outgroup (*Chorthippus pullus*) were collected from natural populations (Appendix A). The specimens were preserved and stored in 96% ethanol at −20 °C for the subsequent molecular genetic investigation. The total DNA was extracted from the hind femora using the AmplySens “DNA-sorb-B” Blood & Tissue kit (InterLabServise, kat. # K1-2-100, Moscow, Russia) according to the manufacturer’s instructions. Each hind femur was cleared of cuticle, dried from the remaining ethanol, and homogenized by a pestle.

### 2.2. DNA Sequencing

The PCR primers for the CR were selected in “Primer3” web version 4.1.0. by using the sequence of *Myrmeleotettix* sp. (MK903595). Three primers were used in this study: MyrmF1 (external forward)—5′-ctggcacgaaatatgccaat-3′, MyrmF800 (inner forward)—5′-cctttttttaactatatggtaaag-3′, and MyrmR2 (external reverse)—5′-gcctctataaacggggtatgaacc-3′. MyrmF1 and MyrmR2 gave a fragment 1076 nucleotides long and it was used for the sequencing of the complete CR. The PCR product of MyrmF800 in combination with MyrmR2 was 691 bp long and contained a partial sequence of the CR and two tRNA genes (*trnI* and *trnQ*).

A PCR amplification of the mitochondrial cytochrome C oxidase subunit I (COI) gene was performed with the universal primers HCO2198 and LCO1490 [30]. A polymerase chain reaction was carried out using the BioMaster HS-Taq PCR-Color (2×) (Biolabmix, kat. # MH010-200, Novosibirsk, Russia) in a volume of 25 μL. The target product was obtained by using a Perkin Elmer GeneAmp PCR System 2400 DNA analyzer. The program included a preliminary denaturation step at 95° for 5 min, 35 amplification cycles, and final elongation at 72° for 7 min. The primer annealing was carried out for 30 s at 60°. The elongation time was 1 min for the MyrmF1–MyrmR2 primer pair and 40 s for the MyrmF800–MyrmR2 primer pair. The COI primer annealing was carried out for 30 s at 48° and the elongation time was 1 min. The PCR products were purified from the reaction mixture using the Cleanup Standard DNA purification kit (Evrogen, kat. # BC022L, Moscow, Russia).

The quality of the PCR products was tested by gel electrophoresis. The sequencing was performed from the PCR primers in both directions with the BigDye Terminator v. 3.1 kit (Thermo Scientific, kat. # 4337454, Waltham, MA, USA) on the genetic analyzer “Applied Biosystems 3500” of the Core Centrum of the Institute of Developmental Biology RAS, according to the manufacturer’s protocol.

The sequences generated for this study were deposited in the NCBI nucleotide database (Appendix A). The data from the BOLD and NCBI nucleotide databases were also used (Appendix A).

### 2.3. Sliding Window Analysis

To reveal the conservative sequence blocks (CSBs) of the CR, we conducted a sliding window analysis of the nucleotide sequence variability. An analysis was performed using a custom Python script based on the equation [31] with a window size of 40 bp and a step size of 1 bp. For the analysis, we used the alignment of the full sequences of the CR in four species of different levels of phylogenetic relationships: *S. eurasius* (PP453567 and PP453568), *Myrmeleotettix* sp. (MK903595), *Pseudochorthippus parallelus* (NC056785), and *Gomphocerippus rufus* (MK903592).

### 2.4. Phylogenetic Analysis

Sequence alignments and the estimation of sequence variability in the CR were performed using the MEGA11 software [32] using the MUSCLE method. The fragment 637–667 bp long including 152 bp of flanking tRNA genes—*trnI* and *trnQ*—was used. In the cases of double peaks caused by the presence of homologous NUMT-derived sequences, we either (1) took the sequences with alternative nucleotide states to analyze if the double peaks did not lead to phylogenetic contradictions, or (2) removed the alternative nucleotide in phylogenetically significant sites if one of the alternative sequences corresponded to a previously identified haplotype, or (3) removed the sequences from the analysis in more complex cases. The degree of divergence within and between the main mitochondrial lineages of Stenobothrini was estimated using a set of unique haplotypes, excluding identical sequences to avoid a decrease in the nucleotide distances between haplotypes. The mean pairwise distances within the groups were estimated for *S. eurasius* and *S. hyalosuperficies*, as well as for the entire set of *Stenobothrus* species. To estimate the degree of divergence between the main mitochondrial lineages, we calculated the mean distances between the groups. The uncorrected *p-distances* and Tamura’s 3-parameter model distances (T3P) were calculated. Tamura’s 3-parameter model corrects for multiple hits, taking into account the differences in the transitional and transversional rates and G+C-content bias. It also assumes an equality of the substitution rates among sites [33]. The mean values of the distances are presented as percentages. To estimate the distances at different levels of divergence, we added two distant taxa, *G. rufus* (MK903592) and *Euchorthippus unicolor* (MK113716).

To estimate the nucleotide diversity (π) within and between *S. eurasius* and *S. hyalosuperficies*, we used all the samples studied, including identical haplotypes (*S. eurasius* from the Altai mountains—23 samples, *S. eurasius* from the contact zone—9 samples, *S. hyalosuperficies* from Kazakhstan—8 samples, and *S. hyalosuperficies* from the contact zone—11 samples) (Appendix A). The analysis was conducted using MEGA11 software [32]. The following parameters were used: *π*, the value of the nucleotide diversity, is the average pairwise difference between all the possible pairs of the individuals in the samples; *π_S_*, the mean value of the nucleotide diversity within the groups, is calculated using the equation 12.72 [34]; *π_T_*, the mean nucleotide diversity of the entire sample, is calculated using the equation 12.73 [34]; *δ_ST_*, the mean interpopulation nucleotide diversity, is estimated as *δ_ST_ = π_T_ − π_S_*; and *N_ST_*, the coefficient of differentiation, is estimated as *N_ST_ = δ_ST_/π_T_*. The standard errors were computed by the bootstrap method. All the values are multiplied by 100.

The phylogenetic trees were reconstructed using the maximum likelihood (ML) method within the phangorn 2.11.1 package (R software 4.2.1). The model of nucleotide evolution was chosen using the “modeltest” function based on the smallest BIC (Bayesian Information Criterion; [35]). The HKY + G(4) + I model was used for the COI reconstruction and TVM + G(4) was used for the CR reconstruction. The bootstrap values were calculated to assess the topology support.

The maximum parsimony (MP) networks of the mitochondrial haplotypes based on the CR sequences were reconstructed in the PopART 1.7 software using the MJN (median joining network) algorithm. If one or several sequences had an ambiguous nucleotide in the parsimony informative position, we changed it to an unambiguous one according to the consensus sequences. Thus, this position was taken into account by the PopART program.

### 2.5. Distribution Map

To illustrate the distribution of the *Stenobothrus* species, we made a map based on our data [10,12,16], the available literature data [7,36,37,38,39,40,41], and the open databases (GBIF.org, Inaturalist.org, and the Orthoptera species file). Unfortunately, the distribution data on many species is incomplete. While the ranges of widespread species, such as *S. lineatus* and *S. nigromaculatus*, are well studied in Europe, their ranges in Asia are almost unknown. Also, *S. carbonarius* is suggested to be a problematic species: despite its wide range (from the south-eastern part of European Russia to Mongolia), findings of this species are very rare. In such difficult cases, we focused on the boundaries of the natural climatic zones [42]. We also show the maximum distribution ranges taking into account a smaller or patchy distribution nowadays because of the biotope disturbance, anthropogenic influences, and climate changes. 

## 3. Results

### 3.1. The Control Region Structure

The CR of *S. eurasius* is located between the 12S rRNA (*rrnS*) and tRNAile (*trnI*) genes (Figure 1), as in most Orthoptera species. The length of the CR in *S. eurasius* is 728 bp. It consists of 84.6% A and T nucleotides and does not contain VNTRs. On the border with the *trnI* gene, there is a long polythymine stretch (T-stretch), polymorphic in length and varying from 7 to 11 bp in different species of the genus *Stenobothrus*. There is another T-stretch found at a distance of 35 bp from the first T-stretch, which is flanked by two purines and is conservative in length (8 bp). To identify CSBs, a sliding window analysis of the nucleotide sequence variability was applied to the alignment of three sequences of the Stenobothrini species (*S. eurasius* and *Myrmeleotettix* sp.) and two sequences of the more distant species (*P. parallelus* and *G. rufus*). The segregating sites are distributed heterogeneously along the region-forming areas with higher density, which alternate with conservative blocks. There are six areas with a low level of polymorphism, which correspond to six CSBs in Orthoptera [43]. There is also a weakly expressed CSB (CSB *x*) situated between CSB E and CSB D (Figure 1).

### 3.2. Phylogenetic Reconstructions and Substitution Rates of the Control Region and the COI Gene

To study the phylogenetic relationships of the tribe Stenobothrini based on mtDNA data, we obtained the sequences of the CR of 14 species including 8 species of the *Stenobothrus* genus and one sequence of *C. pullus* (Gomphocerini) as the outgroup. The total number of samples collected in nature is 70. In addition, sequences from the NCBI nucleotide database (Appendix A) were added to the analysis. The fragment of 691 bp length (full alignment) includes a partial sequence of the CR from *CSB E* to the T-stretch (about 535 bp) and two tRNA genes (*trnI* and *trnQ*) (about 152 bp). The degrees of divergence within and between the main mitochondrial lineages calculated as the mean pairwise distances (p-distance) and Tamura three-parameter (T3P) model distances are shown in Table 1. The values range from 4.89% to 5.34% (p-distance) and from 5.13% to 5.63% (T3P) between *Stenobothrus* and three clades formed by two sister genera, *Myrmeleotettix* and *Omocestus*. These values correspond to the values obtained by the COI gene analysis: from 5.18% to 6.91% (*p-distance*) and from 5.41% to 7.26% (T3P). A comparison of more distant taxa with *Stenobothrus* shows that genetic distances increase from 9.00% (between *Stenobothrus* and *C. pullus*) to 18.48% (between *Stenobothrus* and *Euchorthippus*). At the same time, the distance values for the COI gene are significantly lower (Table 1).

To estimate the substitution rate of the CR and to compare it to the substitution rate of the COI gene, we excluded the tRNA genes from the studied fragment. The distance values of the CR + tRNA genes are slightly lower than that of the CR per se (Table 1). The substitution rates of the CR and COI gene are similar at the level of the *Stenobothrus* and two sister genera comparison. The rate ratios (k) vary in the range of 0.91–1.22. However, the rate ratio increases to 1.83 (between *Stenobothrus* and *G. rufus*) with an increase in the degree of divergence. This indicates that the substitution rate of the COI gene decreases regardless of the linked marker. An independent decrease in the apparent substitution rate of the COI gene with an increase in the divergence degree could be associated with a gradual saturation resulting from multiple substitutions at the same site (homoplasy).

The degree of divergence between haplotypes of the *Stenobothrus* species (Table 2) corresponds to the intraspecific level for *S. eurasius* and *S. hyalosuperficies* (1.03 ± 0.30 for the CR and 1.29 ± 0.28 for the COI gene). Assuming the divergence rate of 3.54% My^−1^ for the COI gene [44], the divergence time of the *Stenobothrus* mt-haplogroup varies from 291 ± 85 ka (CR) to 364 ± 79 ka (COI gene).

To evaluate the resolution capability of the CR as a marker on the level of the Stenobothrini tribe, we compared the topology of the phylogenetic trees based on the CR (Figure 2) and the COI gene (Figure 3). The general topology was found to be the same by the COI gene and the CR reconstructions. We use *P. parallelus* and *C. pullus* as outgroups. The haplotypes of both species occupy the outgroup position, but the order of their branching is not clear. We suggest that the haplotype of *C. pullus* is closer to the Stenobothrini tribe than the haplotype of *P. parallelus*, as is shown by the mean *p-distances* (Table 1).

The first clade of both trees combines the species of *Omocestus*, the *S. stigmaticus* group (*S. stigmaticus* and *S. festivus*), and the species of *Myrmeleotettix*, which corresponds to the previous data [11,17]. Based on the combination of the COI gene and the CR data, five species of *Omocestus* comprise a well-supported monophyletic clade (bootstrap values 97 and 100). Only one species of *Omocestus*, *O. minutus*, is clustered with *M. antennatus* within the COI reconstruction. The latter reconstruction also shows the *S. stigmaticus* group to be a sister group to the *Myrmeleotettix* cluster. The most significant difference between the reconstructions is the topology of *M. palpalis*. According to the COI reconstruction, this species forms a sister group to the *Stenobothrus* cluster, while the CR ML reconstruction suggests monophyly of the *Myrmeleotettix* genus. At the same time, the bootstrap supports of the COI topology of the targeted nodes are higher than within the CR reconstruction.

The main haplogroup (bootstrap values 91 and 92) of both reconstructions included the most *Stenobothrus* species: 8 species of the CR reconstruction and 11 species of the COI reconstruction. This clade has a “star-like” topology and a few clades could only be selected. *S. fischeri* is the only distinguishable species within both trees. The other sequences comprise the group without well-supported clustering (“basal *Stenobothrus* haplogroup”). Some of the S*. nigromaculatus* COI sequences are clustered together and form a poorly supported “*S. nigromaculatus* clade”, but some sequences of other species (*S. lineatus* and *S. cotticus*) are also nested within this clade. The CR sequences of the *S. newskii* and *S. miramae* are clustered together with the low support.

To study the intra- and interspecific variability in *S. eurasius* and *S. hyalosuperficies*, we analyzed the CR of 23 samples of *S. eurasius* from four localities of allopatric populations (Altai), 8 samples of *S. hyalosuperficies* from allopatric populations (Kazakhstan), and 11 samples of *S. hyalosuperficies* and 15 samples of *S. eurasius* from the contact zone (see Appendix A). Most samples of the allopatric *S. eurasius* have their own pattern of synapomorphies, consisting of four parsimony informative substitutions (226 A→T; 238 C→T; 428 C→T; and 464 C→T), and are clustered separately from *S. hyalosuperficies* and *S. eurasius* from the contact zone. The mt-haplotypes of the allopatric and sympatric *S. hyalosuperficies*, as well as of *S. eurasius* from the contact zone, belong to the basal haplogroup of the *Stenobothrus* genus. The values of nucleotide diversity within the groups (*π)* for the allopatric and sympatric *S. eurasius* are 0.17 ± 0.078 and 0.32 ± 0.12, respectively (Table 3), with the mean value of 0.25 ± 0.072 (*π_S_*). However, the mean nucleotide diversity of the entire sample of *S. eurasius* is 0.45 ± 0.13 (*π_T_*). Thus, the mean interpopulation nucleotide diversity (δ*_ST_* = 0.21 ± 0.1) and that one within the population diversity make an almost equal contribution to the nucleotide diversity of *S. eurasius*. The coefficient of differentiation (*N_ST_*) of the allopatric and sympatric populations, which is the proportion (%) of interpopulational diversity [34], is 45.7 ± 15.6. Thus, the populations are well differentiated. Differentiation between the allopatric and sympatric *S. hyalosuperficies* is absent or very low, because the *N_ST_* does not significantly differ from zero. A comparison of the nucleotide diversity of *S. eurasius* from the contact zone with the diversity of *S. hyalosuperficies* indicates very low differentiation (Table 3). Thus, the results show that although the allopatric populations of *S. eurasius* and *S. hyalosuperficies* are well differentiated, there is almost no differentiation between the sympatric populations of these two species. The relationship of the *S. eurasius* haplotypes from the contact zone to *S. hyalosuperficies* indicates a possible asymmetric hybridization between these two species and introgression of the *S. hyalosuperficies* mtDNA into the *S. eurasius* gene pool. One of the samples of *S. hyalosuperficies* (PP239184) from the contact zone still had the haplotype of the Altai *S. eurasius*, while several samples of *S. eurasius* from Altai (PP239140, PP239144, PP239145, and PP239154) had the CR haplotype typical for *S. hyalosuperficies.*

### 3.3. The Network Based on the Control Region

To demonstrate in detail the relationships and clustering of the mtDNA haplotypes, we obtained the MP network using PopART software and the MJN algorithm with modified alignment used for an ML analysis (Figure 4). The Stenobothrini tribe is divided into four haplogroups. The *Omocestus*–*Myrmeleotettix* clade is separated from the *Stenobothrus* clade by 12 phylogenetically significant substitutions. The branching order in the *Omocestus*–*Myrmeleotettix* clade is unclear because of the presence of several homoplasies. It is subdivided into three haplogroups. The first one, the *Omocestus* haplogroup, is separated by 14 phylogenetically significant substitutions. The second one, the *M. maculatus* and *M. antennatus* haplogroup, is supported by eight substitutions. The *M. palpalis* haplogroup is supported by 11 substitutions.

The *Stenobothrus* haplotypes form a separate clade divided into two haplogroups. A “basal *Stenobothrus* haplogroup” was found to be common for all species. Almost all the samples of *S. hyalosuperficies* (except for one sample) from the allopatric populations and the contact zone and all the samples of *S. eurasius* from the contact zone were found to belong to this haplogroup. *S. nigromaculatus* and *S. lineatus* also fall into this clade but do not cluster together. The sequences of the endemic species *S. newskii* (samples from the Altai mountains) and *S. fischeri* (samples from Kazakhstan) are clustered together.

A set of the *S. eurasius* samples from the Altai mountains were found to form the unique haplogroup. One of the *S. hyalosuperficies* samples from the contact zone (PP239184) also belongs to this haplogroup. Surprisingly, the haplotypes of *S. carbonarius* from Kazakhstan are more closely related to the Altaic *S. eurasius* haplogroup than to other *Stenobothrus* species.

## 4. Discussion

### 4.1. Structure and Evolutionary Rate of the mtDNA Control Region

Sequences of the CR for the genus *Stenobothrus* have not been previously published. The structure and localization of the *Stenobothrus* CR are consistent with those previously described for other Orthoptera [43,45,46]. There are six conservative structural elements: CSB A with a T-stretch on the tRnaI border, CSB B, CSB C (TA(A)n-like), CSB D, CSB E (a stem and loop structure), and CSB F (a G+A-rich sequence block). The origin of replication of the major strand is associated with the T-stretch [45,47]. The origin of replication of the minor strand is presumably located in the region of the stem and loop structure (CSB E). It has been shown that the sequence GAA(A)nT flanking the loop structure (in *Stenobothrus n* = 2) is also present in the region of the replication origin of the L-strand in mammals [45,48]. It is also known that the RNA primer for L-strand replication starts to be synthesized on the open part of the loop structure of the template L-strand, when the DNA polymerase replicating the H-strand passes through the loop structure [49]. In invertebrates, this occurs when the first chain is almost 100% synthesized. Other CSBs may be also involved in the replication–transcription regulation or in the mtDNA binding to the mitochondrion inner membrane.

Of particular interest is the divergence rate of the CR in comparison with the substitution rate of other mitochondrial regions. As a rule, 2.3% of sequence divergence per 1 million years is taken as an average substitution rate for mtDNA [50]. This rate is confirmed by the study of the mtDNA evolution of darkling beetles (Coleoptera: Tenebrionidae) using the well-established biogeographic barriers to calibrate the molecular clock [44]. However, there are significant differences between different parts of mtDNA regions. The substitution rate corresponds to 3.54% for the COI gene and to 1.06% for 16S rRNA in darkling beetles [44] and to 4.22% for the CytB gene in tiger beetles [51]. The substitution rate of the CR is significantly higher than the rate of other mtDNA regions in *Drosophila* [26] and vertebrates [52], with significant heterogeneity among the regions and among the sites. Our data show that the substitution rates of the CR and of the COI gene are similar at the level of comparison of *Stenobothrus-Myrmeleotettix-Omocestus* genera (*k* = 0.91–1.22). But the substitution rate of the CR increases compared to the COI gene rate at the split level of *Stenobothrus* and *C. pullus* and of more distantly related outgroup species. The substitution rate could be variable at different levels of evolution, but the evolutionary rates of two linked markers should change in a correlated manner and the rate ratio (*k*) should remain constant. The increase in the rate ratio to 1.83 indicates the decrease in the apparent substitution rate of the COI gene regardless of the CR rate. One of the reasons could be the genetic saturation effect caused by multiple substitutions at the same site (homoplasy), which leads to underestimation of the evolutionary rate and, subsequently, to overestimation of the divergence time. We suppose that the genetic saturation affected the divergence rate of the CR at deeper levels of phylogeny than the rate of the COI gene. We also suggest that the CR reflects the time of divergence at the level of the subfamily Gomphocerinae more accurately than the COI gene.

### 4.2. Phylogenetic Relationships and Taxonomy within the Tribe Stenobothrini

The reconstructions based on both markers (Figure 2 and Figure 3) and the network based on the CR (Figure 4) were shown to have a similar topology. The tribe Stenobothrini appeared to divide into four groups: (1) the *Omocestus* genus, (2) the controversial clade of the species (*M. maculatus, M. antennatus, M. pallidus, S. stigmaticus, S. festivus*, and *O. minutus*) from all three genera, (3) the clade of *M. palpalis*, and (4) the *Stenobothrus* genus. All three genera of Stenobothrini were found to be paraphyletic. Our results mainly support the previous phylogenetic studies of Gomphocerinae [11,17,18,53]. However, all the studies confirm the monophyly of five species of *Omocestus* (*O. viridulus*, *O. rufipes, O. haemorrhoidalis*, *O. Petraeus*, and *O. panteli*) and most species of *Stenobothrus* except for the *S. stigmaticus* group.

At the same time, the topology of the two clades including the *Myrmeleotettix* species is debatable. We suggest that the *S. stigmaticus* group is closer to *Myrmeleotettix* than to *Stenobothrus*, and *O. minutus* is closer to *Myrmeleotettix* than to other species of *Omocestus*, which is also supported by previous studies [11,17,53]. One of the characteristic morphological features in the species of the *Myrmeleotettix* genus are the club-shape antennae tips, which distinguishes them from the species of the genera *Stenobothrus* and *Omocestus* [36]. However, the clustering of some species of *Omocestus* or *Stenobothrus* with the *Myrmeleotettix* species and paraphyly of the *Myrmeleotettix* genus indicate the parallel and independent evolution of the club-shape antennae. The club-shape antennae are usually used during the courtship visual display (e.g., [12,54]). Within the genus *Myrmeleotettix*, the antennae in *M. palpalis* are the least thickened at the ends, which is probably correlated with very weak antennae movements during courtship [55]. According to the phylogeny based on the COI gene, *M. palpalis* forms a separate monophyletic clade sister to the *Stenobothrus* genus (Figure 3). The phylogeny reconstruction based on the CR places *M. palpalis* together with other species of this genus but with rather low bootstrap supports (Figure 2). Thus, we suggest the mtDNA lineage of *M. palpalis* to be the most distantly related to other *Myrmeleotettix* species.

We revealed a very complicated structure of the *Stenobothrus* clade. According to the mtDNA data, there is almost no subdivision into taxa within the *Stenobothrus* genus. Only some species (such as *S. fischeri* or *S. newskii*) are distinguished by mtDNA markers. Other haplotypes do not form species-specific clades showing a star-like topology, and haplotypes of the same mitochondrial lineage can be found in different species (Figure 2, Figure 3 and Figure 4). The genus-specific degree of divergence corresponds to the intraspecific level. It is likely that a common ancestor of the *Stenobothrus* genus shared one mitochondrial haplogroup that evolved in the Pleistocene (1.9–0.6 mya). This haplogroup gave a recent and very fast rise to a complex of the modern *Stenobothrus* species. Thus, the modern species have already been fully formed but the mitochondrial gene pool has not yet split. This is an example of the maximum degree of incomplete lineage sorting. We suggest that the genus *Stenobothrus* appears to represent an unusual evolutionary model, showing a later stage of speciation than most other species. At the same time, an absence of clustering by species within the genus *Stenobothrus* resembles the complexity of the *C. biguttulus* group [18,56]. The main mitochondrial lineages of the *C. biguttulus* group diversified about 506 ka, within the mid-Pleistocene. The nominal species are paraphyletic, which indicates the rapid radiation, incomplete lineage sorting, and/or the extensive gene flow. It is remarkable that most species within the *C. biguttulus* group are also widespread and lack morphological apomorphies but can be easily distinguished by songs.

The two species, *S. eurasius* and *S. hyalosuperficies*, are suggested to be very closely related [10]. We, therefore, expected to find relatively low genetic distances between them. At the same time, we assumed that the resolution capacity of the CR as a phylogenetic marker would be sufficient to detect the differentiation between these species in allopatric localities. Our expectations were partially confirmed: allopatric populations of *S. eurasius* from Altai and Europe [41] form their own mt-haplogroup, whereas allopatric populations of *S. hyalosuperficies* fall into the basal haplogroup of *Stenobothrus.* This indicates that the CR haplotypes of *S. hyalosuperficies* have more in common with the more distant *Stenobothrus* species than with the closely related *S. eurasius*. In the contact zone, *S. eurasius* has mt-haplotypes shared with *S. hyalosuperficies*, which might indicate an introgression of mtDNA from *S. hyalosuperficies* to the *S. eurasius* gene pool. This is in good agreement with our preliminary behavioral experiments with the playback of the songs, in which the *S. hyalosuperficies* females showed a reduced selectivity and high responsiveness to the *S. eurasius* songs [16].

The two species used as the outgroups, *P. parallelus* and *C. pullus*, have special places in the genealogy of the Gomphocerini tribe because of the nuclear–mitochondrial discordance. *Pseudochorthippus parallelus* appears to be closer to Stenobothrini than to Gomphocerini according to the mtDNA data [11,17,18,53], whereas the nuclear markers place these species into the Gomphocerini tribe [18]. *Chorthippus pullus* shows a similar discordance [57]. At the same time, the degrees of divergence of the mitochondrial markers between *Stenobothrus* and *C. pullus* or *P. parallelus* are lower than between *Stenobothrus* and *G. rufus* (Table 1). Thus, the latter species undoubtedly belongs to the tribe Gomphocerini (e.g., [17,18]), whereas the assignment of *C. pullus* and *P. parallelus* to this tribe is debatable.

At the same time, certain limitations should be considered when interpreting our data obtained from the two mtDNA markers. These results reflect the evolution of taxa along only the maternal lineage. However, many studies show that nuclear genes provide a tree topology different from that based on mitochondrial genes (e.g., [18,19]). Moreover, even traditional multigene studies have limited effectiveness in grasshoppers due to the large genome sizes, the abundance of mitochondrial pseudogenes, and mitochondrial haplotype sharing [12,20,28,56,57]. We expect that high-throughput methods such as double-digest restriction site-associated DNA sequencing (ddRADseq) would yield much more information and increase the resolution of the genetic data in this family (e.g., [57]).

Another approach to the study of fast evolution could be an analysis of the genes involved in the adaptations and directional changes in the behavioral and morphological characters. For example, a population genomic scan in three species of the *C. biguttulus* group showed several genes that may be involved in song production and hearing as well as genes involved in other traits, such as food preferences and metabolism [58]. It was also shown in *Drosophila* that the genes *tartan* and *Sox21b* underlie the differences in male genitalia [59,60]. In mosquitoes, STAT genes play an important role in immunity. Some species of *Anopheles* were shown to have a retro-duplicated copy of the STAT gene (STAT-B). The greater amino acid divergence in STAT-B might be indicative of an adaptive evolution, particularly in comparison to the ancestral STAT-A gene [61].

### 4.3. Phylogeography of the Genus Stenobothrus and Possible Scenarios of the Origin of S. eurasius and S. hyalosuperficies

The distribution of the *Stenobothrus* species covers the territory from the Iberian Peninsula to Mongolia and China (Figure 5). Among these species, one can distinguish those with large ranges, such as *S. nigromaculatus, S. fischeri*, and *S. lineatus*. Other species distribute over smaller areas: *S. rubicundulus* and *S. miramae* occur in Europe, and *S. carbonarius* inhabit Asia but are very patchy. In the mountains (the Alps, the Carpathians, the Balkan Mountains, the Caucasus, northern Tien Shan, and the Altai), there are many endemic species, such as *S. cotticus, S. clavatus, S. sviridenkoi, S. newskii*, and others.

*S. eurasius* can be attributed to the widespread species, because its range spreads from eastern Europe and the Balkans to Transbaikalia (Figure 5). There are several subspecies of *S. eurasius* in the Balkans and Anatolia. Moreover, the song analysis revealed that *S. eurasius eurasius*, *S. euraisus bohemicus*, and *S. eurasius clavicornis* could be the same species and *S. eurasius macedonicus* and *S. croaticus* could represent another species [33]. However, they were not revised taxonomically. The range of *S. hyalosuperficies* is smaller than that of *S. eurasius*, spreading from the south-eastern part of European Russia to Kyrgyzstan and north-western China. In the south-eastern part of European Russia (Saratov, Samara, and the Orenburg region) and in Kazakhstan, the southern border of the range of *S. eurasius* overlaps with the northern border of the range of *S. hyalosuperficies*, forming the contact zone.

One can distinguish the two areas of maximal diversity of the *Stenobothrus* species. In one area that covers the Balkans, Anatolia, and the Caucasus, there are more than 13 species. In another area that includes the mountains of Kazakhstan, Middle Asia, and Altai, there are more than 10 species. We do not show several endemic species on the map (Figure 5), which are poorly studied and were not sequenced. In both areas, the numerous glacial refugia were suggested by various authors [62,63,64,65,66].

According to the analysis of the mt-haplotypes, almost all the species of the genus *Stenobothrus* studied share one common haplogroup with almost no fixed differences between the species. We suggest all these species diverged recently and have a limited number of dispersal waves from a common refugium, either in the Balkan–Anatolian region or Central Asia. The divergence time estimates of this genus vary from 0.6 mya [18] to 1.9 mya [17]. We, however, estimate this divergence time as 200–300 ka, which might correspond to two–three glacial–interglacial cycles [67]. It was suggested by Hewitt (1996) [68] that the rates of evolution can be expected to differ between the northern and southern parts of a refugium. The edge northern population extended its range northward during each interglacial period and contracted its range southward during each glacial period. In the southern parts of a refugium, by contrast, the habitats contracted, forcing the isolation of populations to more restricted higher altitudes during the warmer interglacial periods. During the following glacial periods, the range of meadows expanded, and the southern populations increased. According to Hewitt (1996 and 2000) [68,69], the lower genetic diversity would be expected in the north as compared with the south of a refugium. However, glaciations are suggested to be about 10 times longer than interglacials (e.g., [65]). Thus, the southern populations could be in contact during each glacial cycle much longer than in isolation at higher altitudes during each interglacial cycle. This could explain the common mt-haplotype in many *Stenobothrus* species, among which one can find either species with large ranges or endemics in the mountains. The mountain endemics seemed to have not had enough time to form separate mt-haplogroups.

The group of *S. eurasius* is of special interest. The European and Altai populations share one haplotype, which is different from the basic mt-haplogroup of the *Stenobothrus* genus. Notably, among the European populations, only the most northern ones were analyzed [41], which are suggested to represent the nominate subspecies of *S. eurasius* [10]. Unfortunately, the south-European populations represented by *S. eurasius macedonicus, S. eurasius clavicornis*, and *S. croaticus* were not genetically studied. Nevertheless, taking into account the diversity of this group in the Balkans, we suggest that the ancestor of these populations could occur in a separate refugium (e.g., the Balkans) from that of the basic haplogroup. By contrast, *S. hyalosuperficies* very likely occupies the ancestral range (e.g., Kazakhstan), where a common ancestor of at least five species of the *Stenobothrus* genus occurred. Thus, our hypothesis that *S. hyalosuperficies* diverged later than *S. eurasius* is not supported. The hypothesis of mutual monophyly is partly supported, because only the European and Altai populations of *S. eurasius* form a separate cluster on the phylogenetic trees (Figure 2, Figure 3 and Figure 4). It is likely that the edge northern population of the Balkan refugium experienced a few periods of population decline and the subsequent expansion, which was accompanied by the formation of the separate mt-haplogroups. We did not obtain a distinct polyphyletic structure of the phylogenetic trees, but nevertheless, one of the *S. eurasius* specimens from Altai (PP239154, Figure 4) had the *S. hyalosuperficies* haplotype. Therefore, we cannot exclude the ancestral polymorphism by which the two species could share a common gene pool.

## Figures and Tables

**Figure 1 insects-15-00592-f001:**
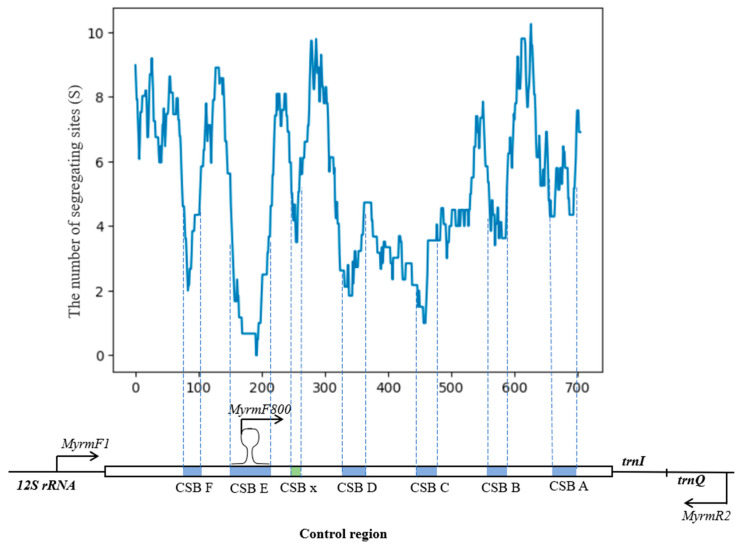
A sliding window analysis of the complete control region on the basis of the sequence alignment in four species of different levels of phylogenetic relationships (*Stenobothrus eurasius, Myrmeleotettix* sp., *Pseudochorthippus parallelus*, and *Gomphocerippus rufus*). The variability plot indicates the number of segregating sites (Ss) in alignment in a window size of 40 bp, with a step size of 1 bp. Blue blocks—the conservative sequence blocks (CSBs) described for Orthoptera [43], green block—the additional CSB *x*, and black arrows—the primers used for the analysis.

**Figure 2 insects-15-00592-f002:**
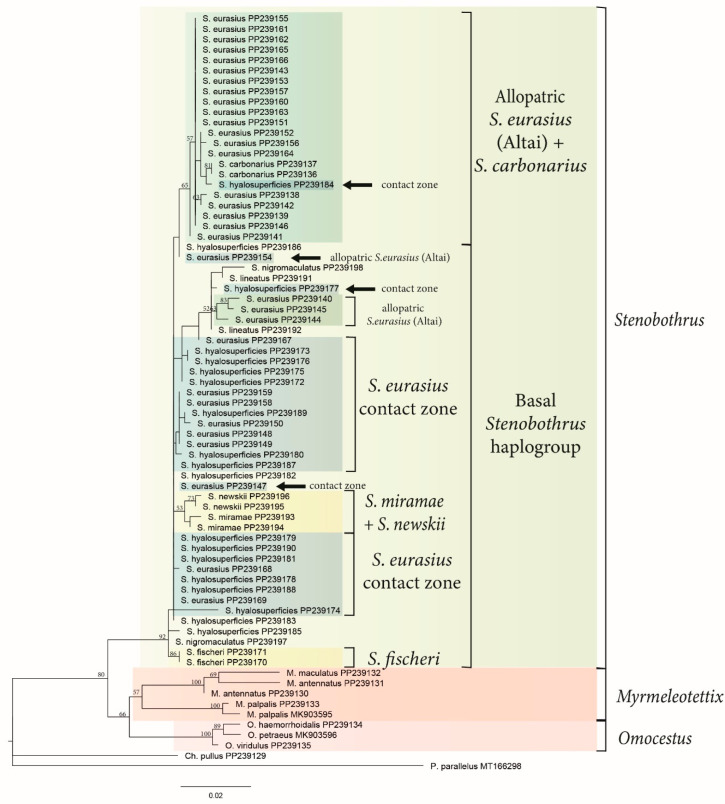
A phylogenetic reconstruction inferred from the maximum likelihood analysis based on the control region (691 bp) of the mtDNA in the Stenobothrini species. Bootstrap values higher than 50 are shown. The monophyletic taxa and other supported clades are marked by color and legends. The scale below the tree indicates the distance.

**Figure 3 insects-15-00592-f003:**
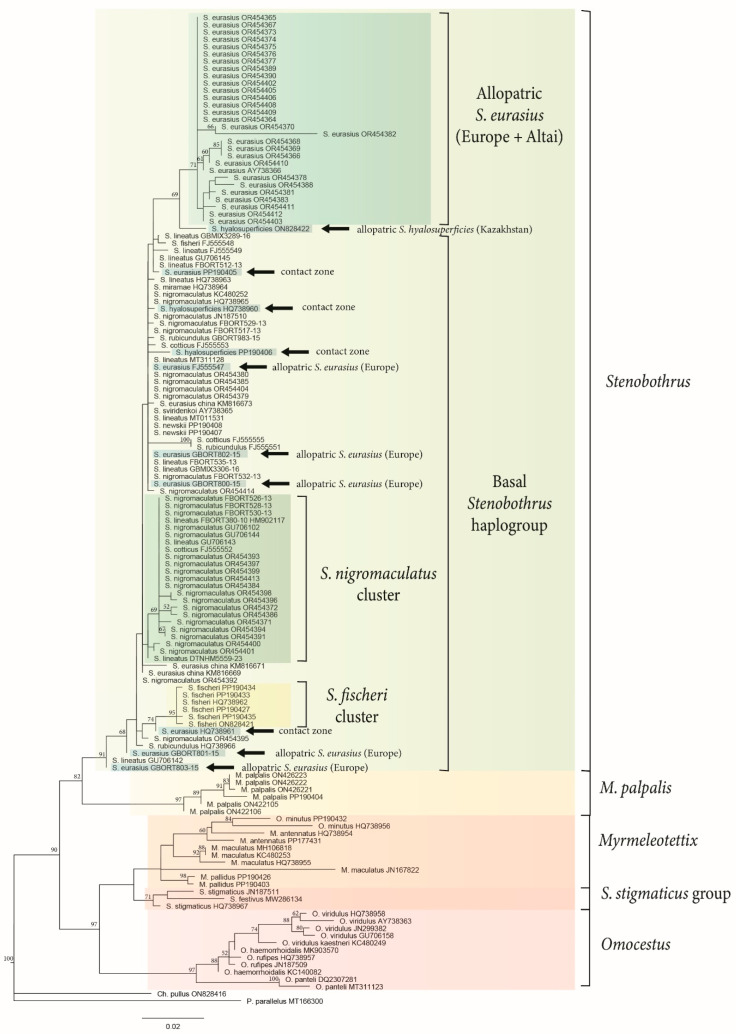
A phylogenetic reconstruction inferred from the maximum likelihood analysis based on the cytochrome oxidase (617 bp) sequences of the mtDNA in the Stenobothrini species. Bootstrap values higher than 50 are shown. The monophyletic taxa and other supported clades are marked by color and legends. The scale below the tree indicates the distance.

**Figure 4 insects-15-00592-f004:**
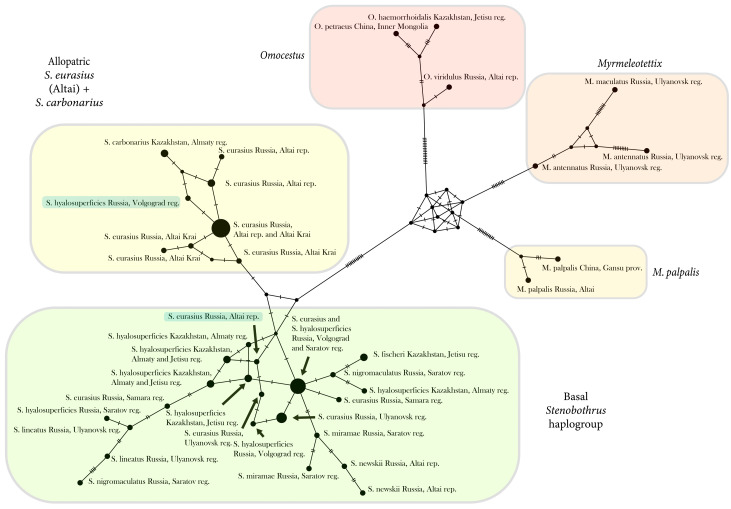
The haplotype network based on the mtDNA control region (691 bp) of the Stenobothrini species. The size of the circles indicates the number of samples with the given haplotype, and the marks on the branches of the network indicate the number of substitutions. The main clades distinguished in this network are marked by color. The *Omocestus*–*Myrmeleotettix* clade is separated from the *Stenobothrus* clade by 12 phylogenetically significant substitutions. The *Stenobothrus* clade is divided into two haplogroups separated by a few substitutions. The *Omocestus*–*Myrmeleotettix* clade is subdivided into three haplogroups separated by 8–14 phylogenetically significant substitutions.

**Figure 5 insects-15-00592-f005:**
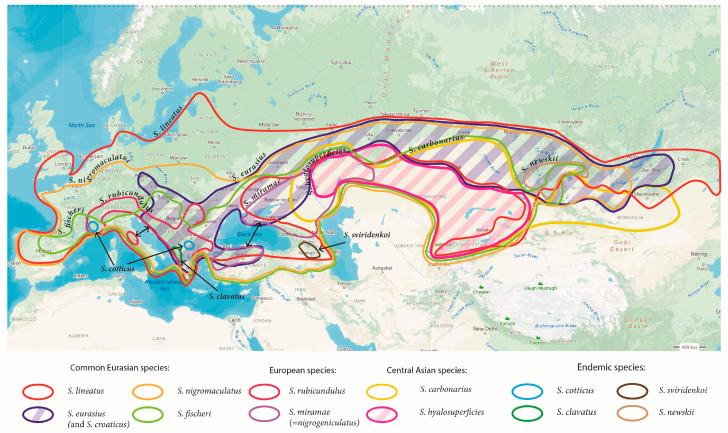
The map of the approximate distribution of the *Stenobothrus* species based on the literature and our own data. We distinguish the species with large ranges (*S. nigromaculatus, S. fischeri, S. lineatus*, and *S. eurasius*), those that distribute over smaller areas (*S. rubicundulus*, *S. miramae, S. carbonarius*, and *S. hyalosuperficies*), and endemic species (*S. cotticus*, *S. clavatus*, *S. sviridenkoi*, and *S. newskii*). The ranges of *S. eurasius* and *S. hyalosuperficies* are highlighted by oblique lines.

**Table 1 insects-15-00592-t001:** The degree of divergence of the control region and the COI gene between the main mitochondrial lineages of Stenobothrini (*Stenobothrus*, *Omocestus*, and *Myrmeleotettix*) and the sister taxa (*Chorthippus pullus*, *Pseudochorthippus parallelus*, *Gomphocerippus rufus*, and *Euchorthippus*).

	Control Region						COI Gene					k CR/COI
	H	S	PIS	p-Distance ± SE	T3P (CR + tRNA) ± SE	T3P (CR)	H	S	PIS	p-Distance ± SE	T3P ± SE	
*Stenobothrus/M. palpalis*	30/2	52	42	4.89 ± 0.79	5.13 ± 0.90	6.58 ± 1.08	41/4	60	43	5.18 ± 0.89	5.41 ± 1.05	1.22
*Stenobothrus/M. maculatus* group *	30/3	64	43	5.34 ± 0.78	5.63 ± 0.87	7.00 ± 1.14	41/9	83	53	6.59 ± 0.88	7.03 ± 1.07	1.00
*Stenobothrus/ * *Omocestus*	30/3	58	45	5.03 ± 0.72	5.26 ± 0.84	6.61 ± 1.12	41/11	82	57	6.91 ± 0.94	7.26± 1.17	0.91
*Stenobothrus/C. pullus*	30/1	72	–	8.24 ± 1.15	9.00 ± 1.38	11.50 ± 1.62	41/1	64	–	6.35 ± 1.06	6.75 ± 1.11	1.7
*Stenobothrus/P. parallelus*	30/1	88	–	11.08 ± 1.20	12.43 ± 1.57	16.50 ± 2.11	41/1	70	–	8.44 ± 1.14	9.13 ± 1.36	1.8
*Stenobothrus/G. rufus*	30/1	94	–	12.56 ± 1.27	14.14 ± 1.70	17.79 ± 2.19	41/1	75	–	8.94 ± 1.12	9.68 ± 1.35	1.83
*Stenobothrus/* *Euchorthippus*	30/1	119	–	15.88 ± 1.47	18.48 ± 1.86	22.84 ± 2.61	41/1	95	–	13.23 ± 1.33	14.75 ± 1.65	1.54

*H*—the number of haplotypes, *S*—the number of segregating sites, *PIS*—parsimony informative sites, *p-distance*—the uncorrected mean pairwise genetic distance (%), T3P—the Tamura 3-parameter model mean distance (%), and SE—the standard error obtained by the bootstrap procedure (100 replicates). CR+tRNA—the partial sequence of the control region and two tRNA genes (*trnI* and *trnQ*), 691 bp; CR—the partial sequence of the control region, 535 bp. * The *M. maculatus* group includes the *M. maculatus*, *M. antennatus, M. pallidus, and O. minutus* sequences of the CR and COI gene.

**Table 2 insects-15-00592-t002:** The characteristics of the nucleotide variability in the control region and the COI gene within the genus *Stenobothrus*.

	Control Region						COI Gene				
	H	S	PIS	p-Distance ± SE	T3P (CR + tRNA) ± SE	T3P (CR)	H	S	PIS	p-Distance ± SE	T3P ± SE
*S. eurasius*	12	10	8	0.59 ± 0.21	0.60 ± 0.21	0.79 ± 0.28	17	35	14	1.47 ± 0.33	1.49 ± 0.28
*S. hyalosuperficies*	8	14	4	0.74 ± 0.22	0.74 ± 0.22	0.99 ± 0.27	3	11	0	1.34 ± 0.42	1.36 ± 0.43
*Stenobothrus*	30	24	17	0.87 ± 0.23	0.88 ± 0.22	1.03 ± 0.3	41	41	22	1.27 ± 0.25	1.29 ± 0.28

*H*—the number of haplotypes, *S*—the number of segregating sites, *PIS*—parsimony informative sites, *p-distance*—the uncorrected mean pairwise genetic distance (%), T3P—the mean pairwise genetic distance using the Tamura 3-parameter model (%), and SE—the standard error obtained by the bootstrap procedure (100 replicates). CR+tRNA—the partial sequence of the control region and two tRNA genes (*trnI* and *trnQ*), 691 bp; CR—the partial sequence of the control region, 535 bp.

**Table 3 insects-15-00592-t003:** The nucleotide diversity within the groups (*π*, on the diagonal), the mean nucleotide diversity for the combined dataset (*π_T_*, above the diagonal), and the coefficients of differentiation (*N_ST_*, below the diagonal) of the allopatric and sympatric populations of *S. eurasius* and *S. hyalosuperficies*. A fragment including 535 bp of the control region and 152 bp of two tRNA genes (*trnI* and *trnQ*) was used.

	*S. eurasius* Altai	*S. eurasius* Contact Zone	*S. hyalosuperficies* Kazakhstan	*S. hyalosuperficies* Contact Zone	*S. hyalosuperficies*
*S. eurasius* Altai	0.17 ± 0.08	0.45 ± 0.13	0.46 ± 0.16	0.43 ± 0.16	0.52 ± 0.17
*S. eurasius* contact zone	45.7 ± 15.6	0.32 ± 0.12	0.39 ± 0.16	0.37 ± 0.11	0.41 ± 0.12
*S. hyalosuperficies* Kazakhstan	45.57 ± 15.63	18.24 ± 9.61	0.32 ± 0.14	0.428 ± 0.123	–
*S. hyalosuperficies* contact zone	32.56 ± 12.96	1.34 ± 3.58	13.82 ± 8.77	0.41 ± 0.13	–
*S. hyalosuperficies*	42.43 ± 11.69	8.57 ± 3.58	–	–	0.43 ± 0.12

All values (mean ± standard error) are multiplied by 100. Standard errors were computed by the bootstrap method.

## Data Availability

All the sequences obtained from this study were deposited in NCBI GenBank under the accession numbers PP239129-PP239198 and PP453567-PP453568 for the CR and PP177426-PP177427, PP177431-PP177435, and PP190403-PP190408 for the COI gene.

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
