# Peer review of "The Fast Evolution of the Stenobothrini Grasshoppers (Orthoptera, Acrididae, and Gomphocerinae) Revealed by an Analysis of the Control Region of mtDNA, with an Emphasis on the Stenobothrus eurasius Group"

_insects, 2024, doi:10.3390/insects15080592_

Round 1
Reviewer 1 Report
Comments and Suggestions for Authors
The authors present a manuscript on the evolution of two closely related grasshopper species. Overall, the manuscript is well-written, the analyses are sound and the conclusions mostly well-supported.
I have two main points of criticism:
1. The authors refer to unpublished data on the behavior of these two species twice, in l. 70 of the introduction and in l. 474 of the discussion. This is not really accessible, so I see three options:
- Reference as a manuscript in preparation, if the plan is to publish the results separately.
- Include them in this manuscript.
Personally, I favor the second option – why not include these results here? Therefore, I suggest the authors to either a) include this part of their research in this manuscript or b) justify why this is not done, and point the reader to where these results can be expected to be found.
2. Basing a phylogenetic study at species level on just two mitochondrial markers is generally valid. However, the severe limitations of this approach have been shown many times (e.g., [17] and https://doi.org/10.1093/biolinnean/blad106). I do not suggest that any additional analyses should be run, but I suggest that it should be made much clearer that these results are based on limited data, and studies based on larger parts of the genome may show different results.
Minor comments:
l. 82: “It is suggested that females of younger…”: Add a reference. Otherwise, if this is a hypothesis that you propose, move it to the discussion.
l. 87: “of taxa to be difficult”: Delete “to be”.
l. 107: “If S. hyalosuperficies diverged from S. eurasius…”: I suggest structuring all three hypotheses in the same way. So: “If the phylogenetic tree shows S. eurasius as paraphyletic with regard to S. hyalosuperficies, this would indicate…”.
l. 130: “Myrmeleotettix sp.”: Un-italicize “sp.”.
Figure 1: The caption describes “three species of different levels of phylogenetic relationships”, but I only see one trajectory. Should I see three?
l. 145: The HCO / LCO primers have been found to be problematic by amplifying more numts in Orthoptera that other, more specific COI primers. I suggest the authors to justify why the think that HCO / LCO are still suitable and describe how they addressed the numt problem.
Table 3: How was Nst calculated? This was not mentioned in the methods section.
l. 228: “unpublished data”: If this is published in this manuscript for the first time, it is rather new data than unpublished data.
l. 249: Italicize “G. rufus”.
l. 260: “691 bp long”: replace “long” with “length”.
l. 300: Insert a period / full stop between “[11, 16]” and “Based”.
Figure 4: This is too small for me to read.
l. 457: Insert a period / full stop between “[17, 50]” and “The main”.
l. 485: Change “accessory” to “the assignment”.
Figure 5: This is also too small for me to read. Although the figure looks really excellent. Except I think S. eurasius does not occur that far North and West in Central Europe.
l. 522: “The edge northern population extended its range towards the north”: This sentence reads somewhat clumsy, please rephrase.
Kind regards
Reviewer 2 Report
Comments and Suggestions for Authors
Dear Author/s,
Manuscript ID: insects-3043595- "Fast evolution of the……Stenobothrus eurasius group" contributes valuable comprehensions to understanding the evolutionary diversification of a control region of mtDNA in Stenobothrini grasshoppers (Orthoptera, Acrididae, Gomphocerinae). The manuscript's (research) is interesting and includes comprehensive genomic data and phylogenetic analysis of orthopteran species and provides a strong foundation for its relationships for fast diverging species. Indeed, this study makes a significant contribution to understanding the significance of evolution of gene and genomes.
Strengths:
1. Relevance and Significance: The study addresses an important and interesting topic, particularly in the context of the gene evolution through divergence. The manuscript's emphasis on the mechanism of evolution derived from purifying selection which is relevant, and its findings contribute to our understanding of the factors that impacted the challenging environment millions of years ago.
2. Comprehensive Data Collection: The authors have employed a comprehensive approach for data collection of genome sequences and annotation, involving a wide variety of models/software to validate its performance. The use of anonymous questions and collateral mechanisms adds robustness to the study's findings.
Suggestions for Improvement:
1. Limitations and Generalizability: Manuscript should acknowledge some limitations, such as the lack of other insect’s species genome data inclusion in your analysis. Although, this would be helpful to check the potential for accuracy of bias on the same phenomenon in other ancient arthropods. Further inclusion in discussion on the applicability and generalizability of the findings to other species would enhance the manuscript's completeness (see minor comment #4 and 5).
2. Literature Review and Context: The manuscript has done comprehensive/exhaustive review of the existing literature on different dynamics. Authors should have presented them in well-verse format. This would help the reader to better understand how the current study contributes to the phylogenetic field and differentiates itself from previous research.
Minor Comments:
1. The title is too long, if possible, make it concise and explicit.
2. Methods of the paper need more description and explaining the default setup of used software would be helpful to other users to replicate the same exercise elsewhere.
3. Supplementary Table S1 title need refurbishing, Bold accession number are not visible (as mentioned), List of “Insect” species should be there.
4. Figure 4, 5 fonts are not visible, make them bigger. The figure should appear serially in the results section. I would suggest, figures and tables provided, including supplementary, are essential for understanding the results. To enhance clarity, consider providing more descriptive figure legends that explain the main findings depicted in each figure.
5. References which do not belong to insect evolution, not good to keep in the list and discussion (For example ref 47). Citing in discussion papers like PMID: 27664587; and PMID: 34578158 would be useful to understand the gene duplication and divergence is central phenomenon in other ancient insect species.
6. References should be registered in a uniform manner, pay attention to write it consistently.
Overall, I would recommend justifying all these minor mistakes at your end. Above are some examples; authors should take care of rest likewise and proof-read (English and Grammar) at your end. Addressing the suggested improvements and major revisions would enhance the manuscript's impact and readability.
All the best.
Comments on the Quality of English LanguageThe manuscript's language and writing style are generally clear and concise. However, some sentences appear to be lengthy and could be restructured for better readability. In the supporting method and discussion section extensive language correction is needed.
Round 2
Reviewer 1 Report
Comments and Suggestions for Authors
The authors revised the manuscript according to the comments. I think that the text has been considerably improved. In my view, citing a conference paper for preliminary results is a suitable solution. All methodologically weak points have been addressed and justified. I have only a few minor comments that should be considered before further processing the manuscript.
l. 195: “S. eurasius и S. hyalosuperficies”: Replace “и “ / “u” with “and”
l. 411: Please check “tiger beetles [51]44.”
l. 477: “closely related species [10]”: Delete “species” (redundancy).
l. 559: Change “haplogroup” to “haplogroups”.
Kind regards
Oliver Hawlitschek
Author Response
Thank you for pointing the minor comments out.
We agree with all of them and have made corrections in the new version of the MS.
Sincerely,
Varvara Vedenina, on behalf of S. Sorokina, N. Sevastianov and T. Tarasova
Reviewer 2 Report
Comments and Suggestions for Authors
I would like to appreciate the efforts of authors for making the necessary amendments to the manuscript and provide reasonable answers to the queries raised by the reviewers. The current version of revised manuscript Insects-3043595-v2 is comparatively looks better now, although I can see there some mistakes still there, which need to be taken care of. For example
Result section 3.3 and method section 2.1 and 2.2 are solely relies on supplementary Table S1. There is a confusion in table S1 between the used database BOLD - Barcode of Life Data System and the bolded accession number in columns, better to highlighted them in a diffrent way (color scheme or underline).
There should be segregation in supplementary table S1 among the sequences dreived from this study (after PCR and sequencing) and what you have arbitrary picked up from the NCBI, including outlier group of insects.
In the section 2.1 and 2.2- have mentioned that some of methods (genetic material isolation and sequencing) has followed using manufacturer’s instruction (line 123 and line 145-148) but catalouge numbers of the molecular biology reagents are missing. It would be helpful, if you provide the product numbers of used reagents.
Authors have replied that they have taken care of reference list, but it has still some irregularities and not been itemized uniformly. Specifically, they have not providing the doi of all articles, and some of the doi is not supporting or may be wrong (example - article no, 6, 10 and 12), hence not get opened.
Choose the relevant and appropriate articles, dont do self-citations, if it not necessary.
Evolutionary divergence and rapid evolution among various insects species have been investigated more commonly by other workers, author should discuss this phenomenon with the help of https://doi.org/10.1016/j.gene.2016.09.022
Refering to an article or book is good, but their equation numbers 12.72 and 12.73 from that reference are not meaningful here (Line #202-2023) better to use the equation.
Author’s has replied that “we would be grateful if the Reviewer points out the specific sentences, which need a correction“. In the revised version itself, there are so many mistakes and not possible to enlist everything here. Althogh here are some example-
Line number 152 and 159 – Analysis vs analyses spelling varies.
Line number 96 – Variable Number of Tandem Repeats (VNTR)
Line number 111 – Evolutionary tree
Line number 127 – „Designed“ not selected
Make sure you have expand all the acronym at least once.
Similarly, you have used italic name of species then it is universally done throughout the manuscript.
There are plenty of other minor mistakes too. Above are some examples, kindly take care and verify the methods, discussion as per the suggestion. Overall, authors need to proofread and present the information in a flawless manner.
All the best.
Comments on the Quality of English LanguageI mentioned some of the language related mistakes in the main comment section too. I will again urge to author that redo the correction of language error, with the help of native English expert/speaker or MDPI’s language service. Overall, this manuscript still needs extensive proofreading.
Author Response
Thank you very much for taking the time to review this manuscript. Please find the detailed responses below.
Comments 1 and 2 about Table S1: Result section 3.3 and method section 2.1 and 2.2 are solely relies on supplementary Table S1. There is a confusion in table S1 between the used database BOLD - Barcode of Life Data System and the bolded accession number in columns, better to highlighted them in a diffrent way (color scheme or underline).
There should be segregation in supplementary table S1 among the sequences dreived from this study (after PCR and sequencing) and what you have arbitrary picked up from the NCBI, including outlier group of insects.
Response: We have highlighted the used database BOLD in different colour and distinguished sequences used by other authors from our sequences.
Comment 3: In the section 2.1 and 2.2- have mentioned that some of methods (genetic material isolation and sequencing) has followed using manufacturer’s instruction (line 123 and line 145-148) but catalouge numbers of the molecular biology reagents are missing. It would be helpful, if you provide the product numbers of used reagents.
Response: We have included the catalogue numbers.
Comment 4: Authors have replied that they have taken care of reference list, but it has still some irregularities and not been itemized uniformly. Specifically, they have not providing the doi of all articles, and some of the doi is not supporting or may be wrong (example - article no, 6, 10 and 12), hence not get opened.
Response: We have provided more doi of the articles, where they are awailable.
Comment 5: Evolutionary divergence and rapid evolution among various insects species have been investigated more commonly by other workers, author should discuss this phenomenon with the help of https://doi.org/10.1016/j.gene.2016.09.022
Response: We have included a paragraph in Discussion (in the end of chapter 4.2), where we discussed other approach to the studies of divergence and rapid evolution among various insect species.
Comment 6: Refering to an article or book is good, but their equation numbers 12.72 and 12.73 from that reference are not meaningful here (Line #202-2023) better to use the equation.
Response: We decided to remain the equation numbers from the reference, similarly to some other authors (e.g., https://academic.oup.com/jeb/article-abstract/18/3/703/7323760?login=true; https://royalsocietypublishing.org/doi/abs/10.1098/rspb.2005.3127; https://www.frontiersin.org/journals/ecology-and-evolution/articles/10.3389/fevo.2017.00143/full; https://academic.oup.com/sysbio/article/59/6/674/1710706?login=true).
Comment 7: Author’s has replied that “we would be grateful if the Reviewer points out the specific sentences, which need a correction“. In the revised version itself, there are so many mistakes and not possible to enlist everything here. Althogh here are some example-
Line number 152 and 159 – Analysis vs analyses spelling varies.
Line number 96 – Variable Number of Tandem Repeats (VNTR)
Line number 111 – Evolutionary tree
Line number 127 – „Designed“ not selected
Response: We have corrected the mistakes mentioned above.
Comments on the Quality of English Language: I mentioned some of the language related mistakes in the main comment section too. I will again urge to author that redo the correction of language error, with the help of native English expert/speaker or MDPI’s language service. Overall, this manuscript still needs extensive proofreading.
Response: We have asked for the help of native English expert in proofreading the text of MS, and some language mistakes have been corrected. In particular, we have much changed language of Abstract.
Sincerely yours,
Varvara Vedenina, on behalf of S. Sorokina, N. Sevastianov and T. Tarasova.
Round 3
Reviewer 2 Report
Comments and Suggestions for Authors
I would like to appreciate the efforts of authors for making the required amendments to the manuscript (insect-3043595-v2) and provide sufficient answers to the queries raised by the reviewers. It is good to have the appropriate references and/or detailed methods, catalogue numbers etc., when you are referring to or establishing any novel ventures.
The current form of the manuscript is looks much better now, although I can see there some minor mistakes still there, which need to be taken care of. For example, line numbers 262-263, 442, 525 it should be mya (Millions of Years Ago) or kya? Or else you can elaborate "kya" if that is correct.
I would like to again urge please take care of these minor mistakes at your end. Proofread the manuscript one more time and try to make it flawless as much as you can.
All the best.
Author Response
Thank you very much for taking the time to review this manuscript.
Comment 1: Line numbers 262-263, 442, 525 it should be mya (Millions of Years Ago) or kya? Or else you can elaborate "kya" if that is correct.
Response: In all lines mentioned in the comment, we meant thousands of years ago. We have corrected kya to ka, since it is a more common abbreviation.
Sincerely yours,
V. Vedenina and co-authors